# Disparities in potential years of life lost due to intimate partner violence: Data from 16 states for 2006–2015

Laurie M. Graham[1]*, Shabbar I. Ranapurwala[2,3‡], Catherine Zimmer[4‡], Rebecca J. Macy[2,5‡], Cynthia F. Rizo[5‡], Paul Lanier[2,5‡], Sandra L. Martin[2,6‡]

1 School of Social Work, University of Maryland-Baltimore, Baltimore, Maryland, United States of America, 2 Injury Prevention Research Center, University of North Carolina at Chapel Hill, Chapel Hill, North Carolina, United States of America, 3 Department of Epidemiology, Gillings School of Global Public Health, University of North Carolina at Chapel Hill, Chapel Hill, North Carolina, United States of America, 4 The Odum Institute, University of North Carolina at Chapel Hill, Chapel Hill, North Carolina, United States of America, 5 School of Social Work, University of North Carolina at Chapel Hill, Chapel Hill, North Carolina, United States of America, 6 Department of Maternal and Child Health, Gillings School of Global Public Health, University of North Carolina at Chapel Hill, Chapel Hill, North Carolina, United States of America

‡ These authors also contributed equally to this work.
* laurie.graham@ssw.umaryland.edu

**Data Availability Statement:** The data underlying the results presented in the study are available from the United States Centers for Disease Control and Prevention (CDC) as part of the National

## Abstract

### Background

Intimate partner violence can lead to deaths of one or both partners and others (i.e., corollary victims). Prior studies do not enumerate the societal cost of intimate partner violence-related fatalities, exclude corollary victims from most analyses, and do not describe groups who bear the highest societal costs from intimate partner violence.

### Objective

We examine racial/ethnic and gender-based disparities in potential years of life lost (PYLL) among intimate partners and corollary victims of intimate partner violence-related mortality.

### Methods

We used 16 US states' 2006–2015 National Violent Death Reporting System data to estimate PYLL among intimate partners ($n = 6,282$) and corollary victims ($n = 1,634$) by victims' race/ethnicity and sex. We describe fatalities by sex, race/ethnicity, age, and victim-suspect relationships and used hierarchical linear models to examine PYLL per death differences by victims' sex and race/ethnicity.

### Results

Nearly 290,000 years of potential life were lost by partner and corollary victims as a result of IPV in 16 states during the decade of study. Most partner victims were female (59%); most corollary victims were male (76%). Female intimate partners died 5.1 years earlier (95% CI: 4.4., 5.9) than males, and female corollary victims died 3.6 years (1.9, 5.5) earlier than

Violence Death Reporting System Restricted Access Database (NVDRS RAD file). Anyone who wishes to use these data must adhere to this same process, which is described here: https://www.cdc.gov/violenceprevention/datasources/nvdrs/RAD.html. The contact for inquiries about these data is: nvdrs-rad@cdc.gov.

**Funding:** The Caroline H. and Thomas S. Royster Fellowship awarded by the University of North Carolina at Chapel Hill Graduate School and the Injury and Violence Prevention Fellowship awarded by the UNC Injury Prevention Research Center supported, in part, LMG's time and effort on this research by supporting her pursuit of a doctoral degree. Funds from both the University of Maryland-Baltimore Open Access Publishing Fund and UNC Injury Prevention Research Center paid for the open access publication fee.

**Competing interests:** The authors have declared that no competing interests exist.

males. Racial/ethnic minorities died nine or more years earlier than their White counterparts. White males had the lowest PYLL per death of all sex/race groups.

## Implications

Intimate partner violence-related fatalities exact a high societal cost, and the burden of that cost is disproportionately high among racial/ethnic minorities. Future interventions targeting specific sex and race/ethnic groups might help reduce disparities in intimate partner violence burden.

## Introduction

Intimate partner violence (IPV) is a significant global issue with numerous detrimental outcomes. IPV is physical/psychological/sexual abuse; threats of such abuse; stalking; and other abusive acts committed within current or former intimate partnerships [1]. IPV-related fatalities include deaths of an intimate partner or others (i.e., corollary victims) resulting from IPV. Prior research mainly addresses IPV burden by describing the incidence of IPV-related fatalities, especially among intimate partner victims of intimate partner homicide (IPH) or intimate partner homicide-suicide (IPH-suicide; [2–5]). This approach fails to quantify the true extent of IPV-related harm by both excluding corollary victims and disallowing the assessment of societal costs of IPV. As such, the current study estimates the societal costs of IPH, IPH-suicides, and fatalities resulting from law enforcement intervention in IPV in terms of potential years of life lost (PYLL) for partners and corollary victims. Findings from such research will help direct scarce IPV prevention resources, both in terms of allocating resources for the delivery of existing prevention strategies and development and testing of new strategies, to those most in need of intervention and protection.

Studies have found that approximately 50% (or more) of female US homicide victims are murdered by intimate partners [2, 6, 7]. For men, this estimate is 5–8% [7, 8]. Limited prior research that examined IPH and IPH-suicide incidents shows that corollary victims account for approximately 20% of IPH victims [9]. Corollary victims include children, family members, and friends/acquaintances of intimate partners, law enforcement officers, and strangers. These people may be killed for intervening to stop IPV or for being the child or new partner of someone in an abusive relationship [9, 10].

Research also suggests that IPV-related fatality rates among male and female partners and corollary victims vary by race/ethnicity [7, 9]. The only multistate study on IPH and IPH-suicide to compare differences in victimization rates for men and women by race/ethnicity found that across 16 states from 2005–2010, non-Latina Black women experienced the highest IPH rate (2.24 per 100,000), followed by Latina women (1.01 per 100,000), Black men (.98 per 100,000), White women (.83 per 100,000), White men (.20 per 100,000), and Latino men (.19 per 100,000; [7]). Hence, racial/ethnic and gender-based differences must be considered to inform targeted IPV prevention efforts.

IPV-related fatality rates are a critical measure of incidence, but they do not characterize the societal costs of IPV-related fatalities. PYLL, a measure of the number of years individuals may have lived if their life had not ended prematurely because of a specific cause, is one estimate of lost societal potential due to IPV [11–13]. Unlike rates and risk estimates, PYLL directly enumerates disparities in life expectancy. With such information, research and

interventions that target PYLL disparities will be better equipped to address racial/ethnic gaps in life expectancy.

In this study, we use data from the National Violent Death Reporting System (NVDRS) to examine the burden of IPHs, IPH-suicides, and legal intervention fatalities among male and female partners and corollary victims among various racial/ethnic groups by measuring: (1) total PYLL due to IPV and PYLL for specific sex-race groups; (2) PYLL per IPV-related fatality by sex-race group; and (3) differences in PYLL per IPV-related fatality between sex-race groups.

## Methods

### Data sources

We analyzed NVDRS data [14]. NVDRS is a national surveillance system that abstracts violent death information from death certificates, medical examiner, law enforcement, and toxicology reports. We used data from the 16 states (Alaska, Colorado, Georgia, Kentucky, Maryland, Massachusetts, New Jersey, New Mexico, North Carolina, Oklahoma, Oregon, Rhode Island, South Carolina, Utah, Virginia, and Wisconsin) that contributed data every year from 2006–2015. We also used National Center for Health Statistics data on the average US life expectancy [15]. The Institutional Review Board (IRB) at the University of North Carolina-Chapel Hill deemed this non-human subjects research.

### Sample selection

Guided by prior research [9, 10], we identified NVDRS homicides, suicides, and legal intervention deaths resulting from IPV, jealousy, and/or an intimate partner problem using NVDRS coded variables [16]. All cases endorsed for IPV, including those with incident identification numbers connected to cases endorsed for IPV, were included in the sample. A study team member conducted narrative reviews for all cases endorsed in the NVDRS for "intimate partner problem" and/or "jealousy" but not IPV to determine whether the incident met the definition of IPV and should thus be included in the study sample. For each narrative review, the team member read available medical examiner and law enforcement narratives to determine if the case met the NVDRS definition of IPV (and could thus be considered IPV-related): "…physical violence, sexual violence, stalking and psychological aggression (including coercive acts) by a current or former intimate partner" [17, para 1]. Fig 1 provides details of sample selection.

Using NVDRS's mutually exclusive victim-suspect relationship categories, we identified ex-girlfriend/ex-boyfriend, ex-spouse, current girlfriend/boyfriend, current spouse, or boyfriend/girlfriend unspecified as current/former (intimate partners) and all other relationship categories (corollary victims). Thus, the study sample included: homicides in which the victim was killed by an intimate partner or killed during an IPV incident; homicides resulting from law enforcement intervention in an IPV incident (legal intervention deaths); and suicides that occurred after killing an intimate partner. NVDRS lacks information on non-binary gender identities (pre-2013). Hence, the study was limited to male and female victims [16]. We excluded unintentional deaths and deaths of undetermined intent [16]. We also excluded incidents with missing information for victim's age, victim-suspect relationship, or suspect information.

The sample included 7,916 IPV-related fatalities (partner: 6,282; corollary: 1,634). Due to small sample sizes and ambiguity of categorization, we excluded racial/ethnic categories of non-Latinx other, two or more races, and unknown races from PYLL estimates for multivariate analyses, reducing the samples (partner: 5,965; corollary: 1,585).

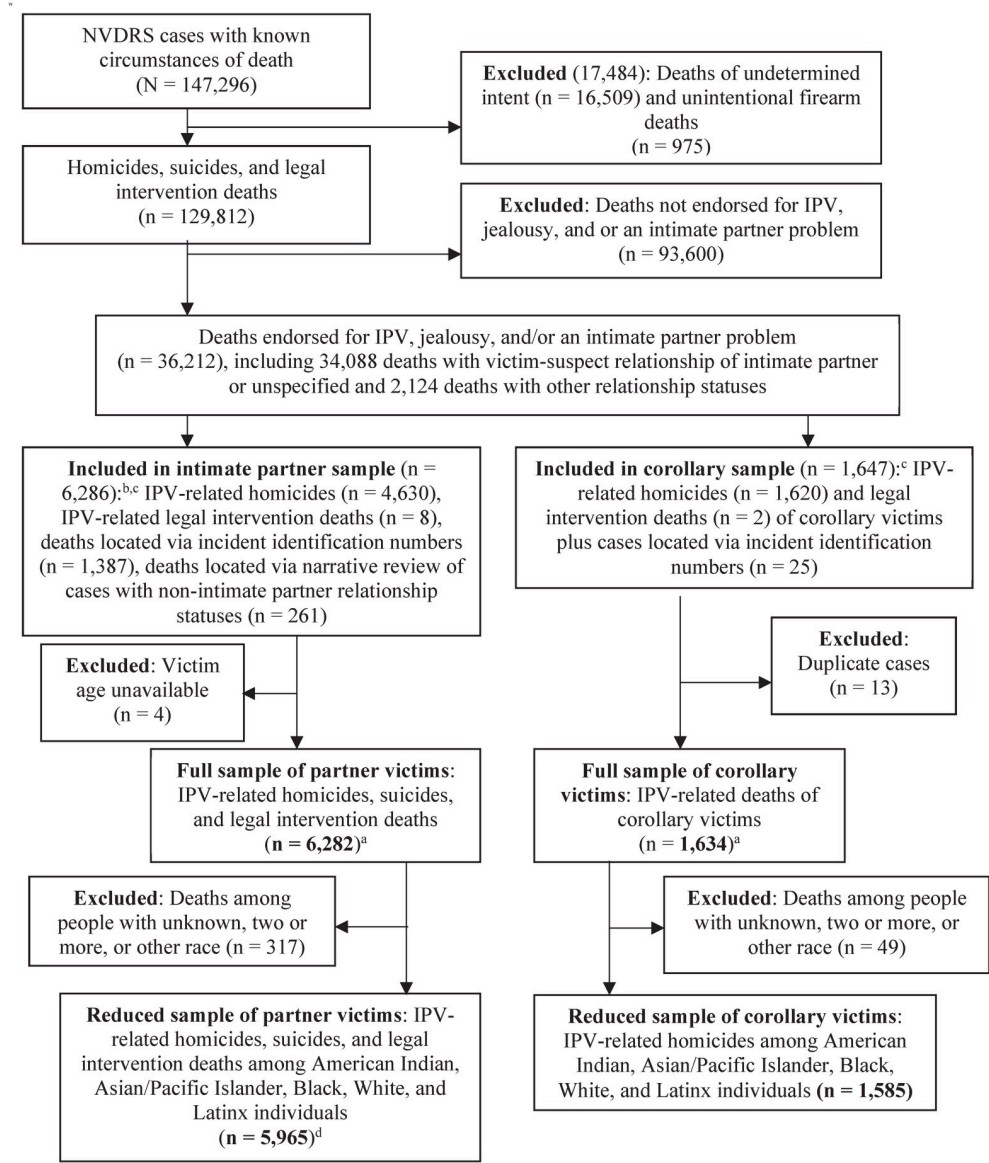

**Fig 1. Flowchart of the sample selection process.** PYLL = potential years of life lost. IPV = intimate partner violence. IPP = intimate partner problem. [a] This sample was used to calculate PYLL due to IPV across all NVDRS racial/ethnic groups. [b] A few deaths in the final sample were inappropriately endorsed for unknown circumstances of death and thus not in the original universe of cases (*N* = 147,296). We located these deaths using incident identification numbers. [c] We read medical examiner and law enforcement reports for all cases endorsed for IPV but without a specified victim-suspect relationship and all cases endorsed for IPP and/or jealousy but not IPV to determined eligibility. [d] This reduced sample was used in multivariate models and estimates of PYLL per death differences among groups.

## Measures

The primary outcome was PYLL due to IPV. We calculated PYLL due to IPV by subtracting age at death from a standard life expectancy value of 75 for all victims [15]. Per established methods [12, 13], victims who died at an age older than the standard life expectancy were given a PYLL of zero. We calculated PYLL per fatality by dividing the total PYLL by the number of fatalities.

Covariates included the death year (calendar years 2006–2015 as a categorical variable) to account for time trends, victim race/ethnicity (American Indian/Alaskan Native, Asian/Pacific Islander, Black, White, Latinx), and victim sex (male, female).

## Data analysis

We used descriptive statistics to describe the study population (victim sex, race/ethnicity, and age; incident type; and victim-suspect relationship). We calculated PYLL and PYLL per fatality by victim's sex and race/ethnicity separately for partner and corollary victims. To examine differences in PYLL per IPV-related fatality among male and female victims by race/ethnicity separately for partner and corollary victims, we used hierarchical linear models (HLM) with maximum likelihood estimation to account for clustering of deaths within the 16 states. Models estimated differences in PYLL per fatality among male and female IPV-related fatalities by race/ethnicity using an interaction term between sex and race/ethnicity. We fit random intercept models to calculate coefficients and 95% Confidence Intervals (CIs). We used a Bonferroni correction for multiple comparisons with the same data [18]. We conducted sensitivity analyses of HLM findings by dropping cases related to an intimate partner problem only as defined by the NVDRS and comparing these findings to findings for the reduced samples used in multivariate models. We completed all data analysis in Stata version 15 in 2018–2019.

## Results

### Demographic characteristics

The demographic distributions of IPV-related fatalities in the full (partner: 6,282; corollary: 1,634) and reduced datasets (partner: 5,965; corollary: 1,585) were similar (Table 1). Most partner victims in the reduced dataset were female and white, followed by Black and Latinx, with an average age of 42.0 years (SD = 15.3; Median = 40; range = 13–100). Most corollary victims in the reduced dataset were male and White, followed by Black and Latinx, with an average age of 32.1 years (SD = 15.9; Median = 31; range = 0–86).

Most partners died by homicide (75.6%), followed by suicide (21.8%) and legal intervention (2.6%). Most partner victims were current spouses (31.5%) or girlfriends/boyfriends (30.4%) of the suspected perpetrator, followed by ex-girlfriends/boyfriends (6.9%), ex-spouses (3.1%), and girlfriends/boyfriends unspecified (2.9%; Table 2). All corollary victims died by homicide. Corollary victim-suspect relationships (Table 2) are defined from the perspective of the victim (e.g., "child" means victim was a child of the suspect). Most victims knew the suspected perpetrator in some manner (33.0%). The next most common relationships were that of an acquaintance (17.7%), child (9.4%), friend (5.0%), and stranger (4.8%). Nine (0.6%) were law enforcement officers dying while intervening in IPV incidents.

### PYLL due to IPV

Combined, all groups lost approximately 289,135 years of potential life from 2006–2015 (partners: 209,636; corollaries: 70,499).

Most partner victims were female. Hence, female victims lost more years of potential life than male victims (128,210 versus 81,426). Similarly, White partner victims lost the highest number of PYLL (96,175 years), followed by Black victims (68,666 years) and Latinx victims (24,173 years). Female partner victims represent a larger percentage of IPV-related fatalities and lost more PYLL than male partner victims across all race/ethnicity groups.

Most corollary victims were male. Hence, male victims lost more potential years of life than female victims (52,665 versus 17,834). Similarly, White corollary victims lost the greatest

**Table 1. Sample characteristics of victims of IPV-related fatalities.**

| | Partner victims | | | | Corollary victims | | | |
| | Full dataset (n = 6,282) | | Reduced dataset (n = 5,965) | | Full dataset (n = 1,634) | | Reduced dataset (n = 1,585) | |
| Characteristic | n | % | n | % | n | % | n | % |
|---|---|---|---|---|---|---|---|---|
| **Sex** | | | | | | | | |
| Female | 3629 | 57.8 | 3503 | 58.7 | 386 | 23.6 | 375 | 23.7 |
| Male | 2653 | 42.2 | 2462 | 41.3 | 1248 | 76.4 | 1210 | 76.3 |
| **Race/ethnicity**[a] | | | | | | | | |
| American Indian/Alaskan Native | 129 | 2.1 | 129 | 2.2 | 39 | 2.4 | 39 | 2.5 |
| Asian/Pacific Islander | 136 | 2.2 | 136 | 2.3 | 29 | 1.8 | 29 | 1.8 |
| Black | 1794 | 28.6 | 1794 | 30.1 | 597 | 36.5 | 597 | 37.7 |
| Latinx | 632 | 10.1 | 632 | 10.6 | 195 | 11.9 | 195 | 12.3 |
| White | 3274 | 52.1 | 3274 | 54.9 | 725 | 44.4 | 725 | 45.7 |
| Other | 23 | 0.4 | -- | -- | 6 | 0.4 | -- | -- |
| Two or more races | 176 | 2.8 | -- | -- | 43 | 2.6 | -- | -- |
| Unknown race | 118 | 1.9 | -- | -- | -- | -- | -- | -- |
| **Age** (years) | | | | | | | | |
| 0–4 | -- | -- | -- | -- | 96 | 5.9 | 96 | 6.1 |
| 5–9 | -- | -- | -- | -- | 54 | 3.3 | 54 | 3.4 |
| 10–17 | 62 | 1.0 | 58 | 1.0 | 83 | 5.1 | 81 | 5.1 |
| 18–29 | 1406 | 22.4 | 1333 | 22.3 | 517 | 31.6 | 500 | 31.5 |
| 30–39 | 1562 | 24.9 | 1477 | 24.8 | 363 | 26.6 | 354 | 22.3 |
| 40–49 | 1521 | 24.2 | 1437 | 24.1 | 287 | 17.6 | 282 | 17.8 |
| 50–59 | 943 | 15.0 | 902 | 15.1 | 150 | 9.2 | 147 | 9.3 |
| 60–69 | 419 | 6.7 | 399 | 6.7 | 49 | 3.0 | 49 | 3.1 |
| 70–79 | 212 | 3.4 | 206 | 3.5 | 15 | 0.9 | 15 | 0.9 |
| 80–89 | 142 | 2.3 | 138 | 2.3 | 7 | 0.4 | 7 | 0.4 |
| 90–99 | 14 | 0.2 | 14 | 0.2 | -- | -- | -- | -- |
| 100 | 1 | 0.0 | 1 | 0.0 | -- | -- | -- | -- |

[a] non-Latinx unless specified.

number of PYLL (29,122), followed by Black victims (26,454) and Latinx victims (9,389). Male corollary victims represent a larger percentage of IPV-related fatalities and lost more PYLL than female corollary victims across all race/ethnicity groups.

## PYLL per IPV death

PYLL per IPV fatality was 35.2 and 30.5 for female and male partner victims, respectively, and 45.7 and 42.1 for female and male corollary victims, respectively. Hence, for partners, female victims overall died younger than male victims.

Across racial/ethnic groups, PYLL per death estimates ranged from 29.4–38.6 for partner victims and 40.2–48.1 for corollary victims. American Indian/Alaskan Native victims died the youngest among partner victims (PYLL per death of 38.6), followed closely by Black (38.3) and Latinx (38.2) partner victims (Table 3). Latinx victims died the youngest among corollary victims (PYLL per fatality of 48.1), followed by Asian/Pacific Islander (46.6), Black (44.3), and American Indian/Alaskan Native (44.1) corollary victims. Considering both sex and race/ethnicity among partner victims, PYLL per fatality ranged from 25.4 years for White male partner victims to 40.6 years for Latinx female partner victims (Table 3). For corollary victims, PYLL

**Table 2. Victim-suspect relationship by intimate partner fatality victim type.**

| | Partner Victims | | | | | Corollary Victims | | | |
| | Full dataset (n = 6,282) | | Reduced dataset (n = 5,965) | | | | Full dataset (n = 1,634) | | Reduced dataset (n = 1,585) | |
| Victim-suspect relationship[a] | N | % | n | % | Victim-suspect relationship[a] | N | % | n | % |
|---|---|---|---|---|---|---|---|---|---|
| Spouse | 1928 | 30.7 | 1876 | 31.5 | Other person known by victim[f] | 528 | 32.3 | 518 | 33.0 |
| Girlfriend or boyfriend | 1863 | 29.7 | 1812 | 30.4 | Acquaintance | 284 | 17.4 | 278 | 17.7 |
| Self-inflicted | 1350 | 21.5 | 1300 | 21.8 | Child | 157 | 9.6 | 148 | 9.4 |
| Ex-girlfriend or ex-boyfriend | 435 | 6.9 | 414 | 6.9 | Friend | 82 | 5.0 | 79 | 5.0 |
| Law enforcement officer[b] | 275 | 4.4 | 154 | 2.6 | Stranger | 76 | 4.7 | 76 | 4.8 |
| Ex-spouse | 188 | 3.0 | 182 | 3.1 | In-law | 57 | 3.5 | 56 | 3.6 |
| Girlfriend or boyfriend, unspecified[c] | 183 | 2.9 | 170 | 2.9 | Child of suspect's girlfriend/boyfriend | 51 | 3.1 | 43 | 2.7 |
| Other intervention led to death[d] | 29 | 0.5 | 27 | 0.5 | Stepchild | 40 | 2.5 | 38 | 2.4 |
| Intimate partner, unspecified[e] | 8 | 0.1 | 8 | 0.1 | Roommate | 25 | 1.5 | 25 | 1.6 |
| Unknown | 23 | 0.4 | 22 | 0.4 | Intimate partner of suspect's parent | 25 | 1.5 | 24 | 1.5 |
| | | | | | Parent | 24 | 1.5 | 24 | 1.5 |
| | | | | | Other family member[g] | 21 | 1.3 | 19 | 1.2 |
| | | | | | Sibling | 19 | 1.2 | 18 | 1.2 |
| | | | | | Stepparent | 16 | 1.0 | 15 | 1.0 |
| | | | | | Victim was law enforcement officer | 9 | 0.6 | 9 | 0.6 |
| | | | | | Other relationship[h] | 15 | 1.0 | 14 | 0.9 |
| | | | | | Unknown | 205 | 12.5 | 187 | 11.9 |

[a] Categories are from the NVDRS and are mutually exclusive.

[b] A law enforcement officer caused the fatality.

[c] Unknown whether current or former girlfriend or boyfriend.

[d] Killed by someone other than self, law enforcement, or partner (e.g., acquaintance, in-law, stranger).

[e] Unknown whether intimate partner was current or former girlfriend, boyfriend, or spouse.

[f] The suspected perpetrator was known by the corollary victim, though the exact relationship was unable to be determined based on NVDRS source documents.

[g] This category refers to family members that do not fit into one of the other specified victim-suspect relationship categories (e.g., cousin, uncle).

[h] This category includes current/former co-worker, rival gang member, grandchild, babysitter, and schoolmate.

per fatality ranged from 37.3 for Asian/Pacific Islander male victims to 53.1 for Asian/Pacific Islander female victims (Table 4).

## Differences in PYLL per IPV fatality

PYLL per fatality differences indicate how much earlier—or later—in life a specific group of individuals died in comparison to another group (e.g., women versus men). Adjusted PYLL per fatality differences indicate that, on average, a female partner victim died about 5 years (difference = 5.1; 95% CI: 4.4, 5.9) younger than a male partner victim. Compared to White partner victims, on average, Black partner victims died 9.5 (95% CI: 8.2, 10.7) years younger, American Indian/Alaskan Native partner victims died 9.0 (95% CI: 5.3, 12.7) years younger, and Latinx partner victims died 9.9 (95% CI: 8.2, 11.7) years younger due to IPV.

On average, a female corollary victim died about 3.6 years (difference = 3.6; 95% CI: 1.9, 5.5) younger than a male corollary victim. Further, compared to White corollary victims, on average, Latinx corollary victims died 8.0 (95% CI: 4.4, 11.5) years younger, and Black corollary victims died 4.3 (95% CI: 1.9, 6.7) years younger.

Among both partner and corollary victims, the relationship between sex and PYLL was modified by race/ethnicity (Tables 3 and 4). As compared to White male partner victims, on

**Table 3. Differences in PYLL per partner IPV-related fatality by sex and race/ethnicity (n = 5,965).**

| Demographic characteristic | n | %[b] | PYLL per death | Unadjusted PYLL per death differences[c] (95% CI) | Adjusted PYLL per death differences[c,d] (95%CI) |
|---|---|---|---|---|---|
| **Sex** | | | | | |
| Female | 3503 | 58.7 | 35.2 | 4.8 (4.0–5.5)*** | 5.1 (4.4–5.9)*** |
| Male (reference) | 2462 | 41.3 | 30.5 | -- | -- |
| **Race/ethnicity** | | | | | |
| AI/AN | 129 | 2.2 | 38.6 | 8.7 (5.0–12.5)*** | 9.0 (5.3–12.7)*** |
| Asian/PI | 136 | 2.3 | 34.1 | 5.0 (1.5–8.6)*** | 5.2 (1.7–8.6)*** |
| Black | 1794 | 30.1 | 38.3 | 9.2 (8.0–10.5)*** | 9.5 (8.2–10.7)*** |
| Latinx | 632 | 10.6 | 38.2 | 9.8 (8.0–11.6)*** | 9.9 (8.2–11.7)*** |
| White (reference) | 3274 | 54.9 | 29.4 | -- | -- |
| **Among female intimate partners[a]** | | | | | |
| AI/AN | 72 | 1.2 | 39.5 | -- | 13.6 (7.8–19.3)*** |
| Asian/PI | 79 | 1.3 | 35.9 | -- | 11.0 (5.7–16.4)*** |
| Black | 1010 | 16.9 | 39.3 | -- | 14.4 (12.4–16.3)*** |
| Latina | 364 | 6.1 | 40.6 | -- | 16.2 (13.5–19.0)*** |
| White | 1978 | 33.2 | 32.0 | -- | 6.7 (5.1–8.4)*** |
| **Among male intimate partners[a]** | | | | | |
| AI/AN | 57 | 1.0 | 37.4 | -- | 11.8 (5.5–18.1)*** |
| Asian/PI | 57 | 1.0 | 31.4 | -- | 6.5 (0.2–12.7)** |
| Black | 784 | 13.1 | 36.9 | -- | 11.9 (9.8–14.0)*** |
| Latino | 268 | 4.5 | 35.1 | -- | 10.7 (7.5–13.8)*** |
| White (reference) | 1296 | 21.7 | 25.4 | -- | -- |

*Note*. PYLL = Potential years of life lost; AI = American Indian; AN = Alaskan Native; PI = Pacific Islander; CI = confidence interval.

**p ≤ .01

***p ≤ .001.

[a] non-Latinx unless specified.

[b] Percent of reduced sample (*n* = 5,965).

[c] A Bonferroni correction was used for calculations with multiple comparisons among groups using the same data.

[d] PYLL per death differences by sex and by race/ethnicity are adjusted for sex, race/ethnicity, and death year and account for state-level clustering within 16 states. PYLL per death differences by both sex and race/ethnicity are adjusted for each of these same variables plus a modification effect (sex*race) and account for state-level clustering within 16 states.

average, Latinx female partner victims died 16.2 years (95% CI: 13.5, 19.0) younger. Similarly, other racial/ethnic and sex groups of partner victims died younger than White male partner victims (Table 3).

As compared to White male corollary victims, on average, a Latinx female corollary victim died 12.1 years (95% CI: 3.6, 20.6) younger, and a Black female corollary victim died 8.8 years (95% CI: 4.1, 13.6) younger. Among male corollary victims, Asian/Pacific Islander victims died 13.9 years (95% CI: 1.6, 26.2), Latinx victims 8.0 years (95% CI: 3.5, 12.6), and Black victims 4.2 years (95% CI: 1.0, 7.4) younger than White male corollary victims.

Sensitivity analyses excluding cases related to an intimate partner problem produced similar results (not shown).

## Discussion

Our study investigated the burden of IPV-related fatalities on various groups of both partner and corollary victims in the United States. We found that nearly 290,000 potential years of life

**Table 4. Differences in PYLL per corollary IPV-related fatality by sex and race/ethnicity (n = 1,585).**

| Demographic characteristic | n | %[b] | PYLL per death | Unadjusted PYLL per fatality differences[c] (95% CI) | Adjusted PYLL per fatality differences[c,d] (95% CI) |
|---|---|---|---|---|---|
| **Sex** | | | | | |
| Female | 375 | 23.7 | 45.7 | 3.5 (1.6–5.3)*** | 3.6 (1.9–5.5)*** |
| Male (reference) | 1210 | 76.3 | 42.1 | -- | -- |
| **Race/ethnicity** | | | | | |
| AI/AN | 39 | 2.5 | 44.1 | 3.9 (3.3–11.1) | 3.7 (3.4–10.8) |
| Asian/PI | 29 | 1.8 | 46.6 | 6.3 (2.0–14.6) | 5.8 (2.4–14.0) |
| Black | 597 | 37.7 | 44.3 | 4.3 (1.9–6.8)*** | 4.3 (1.9–6.7)*** |
| Latinx | 195 | 12.3 | 48.1 | 8.1 (4.5–11.6)*** | 8.0 (4.4–11.5)*** |
| White (reference) | 725 | 45.7 | 40.2 | -- | -- |
| **Among female corollary victims[a]** | | | | | |
| AI/AN | 6 | 0.4 | 38.3 | -- | -1.5 (-22.0–19.0) |
| Asian/PI | 12 | 0.8 | 37.3 | -- | -1.6 (-16.2–13.0) |
| Black | 140 | 8.8 | 47.9 | -- | 8.8 (4.1–13.6)*** |
| Latina | 37 | 2.3 | 52.0 | -- | 12.1 (3.6–20.6)*** |
| White | 180 | 11.4 | 43.5 | -- | 4.3 (-0.01–8.6) |
| **Among male corollary victims[a]** | | | | | |
| AI/AN | 33 | 2.1 | 45.1 | -- | 5.5 (3.4–14.5) |
| Asian/PI | 17 | 1.1 | 53.1 | -- | 13.9 (1.6–26.2)* |
| Black | 457 | 28.8 | 43.2 | -- | 4.2 (1.0–7.4)*** |
| Latino | 158 | 10.0 | 47.2 | -- | 8.0 (3.5–12.6)*** |
| White (reference) | 545 | 34.4 | 39.1 | -- | -- |

*Note.* PYLL = Potential years of life lost; AI = American Indian; AN = Alaskan Native; PI = Pacific Islander; CI = confidence interval.

*$p \leq .05$

***$p \leq .001$.

[a] non-Latinx unless specified.

[b] Percent of reduced sample (*n* = 1,585).

[c] A Bonferroni correction was used for calculations with multiple comparisons among groups using the same data.

[d] PYLL per death differences by sex and by race/ethnicity are adjusted for sex, race/ethnicity, and death year and account for clustering of the deaths within 16 states. PYLL per death differences by both sex and race/ethnicity are adjusted for each of these same variables plus a modification effect (sex*race) and account for clustering within 16 states.

were lost due to IPV from 2006–2015 across 16 states, with some groups experiencing a greater PYLL burden than others.

Over half of the intimate partner victims were female (57.8%), which is consistent with prior research indicating a greater proportion of IPH and IPH-suicide partner victims are female compared to male [9, 19]. Female partner victims also died five years younger, on average, than male partner victims. Differences in PYLL per IPV fatality among partners showed that White men had the lowest average PYLL per death.

The majority of corollary victims were male. Among corollary victims, Black and White men lost the most potential for societal contribution due to IPV. Many corollary victims were new male partners of abused women or men having an affair with the suspect's current female partner, and some male corollary victims were family members of a victim of IPV who intervened to stop violence [9]. Echoing prior research [9], corollary victims had a variety of relationships with suspects, and 1 in 7 corollary victims were ages 17 and younger. The fatalities of young people account for many of the estimated PYLL among corollary victims and greatly

increase the average PYLL per IPV fatality found for various groups. Protecting young people and meeting their unique needs must be a significant focus in IPV-related fatality prevention.

Disparities in PYLL per IPV fatality among partners and corollary victims are likely due to many complex issues. Some differences can result from variations in the age at which different sex/race groups first engage in intimate relationships and the age at first IPV victimization [1, 20]. Nationally, median age of men at first marriage was roughly two years older than that of women from 2006–2015 [21]. Other factors that likely help explain between-group differences include systematic inequities in delivery of and access to critical support services (e.g., advocacy, crisis intervention, legal, or other services) for IPV survivors and their loved ones and differences in IPV incidence rates across groups.

Prior IPV is the strongest risk factor for being killed by an intimate partner [22]. Hence, learning how to intervene with IPV and stop violence before it turns lethal is of utmost importance. Relative to others, some groups lack access to culturally and linguistically appropriate IPV support services [23–25]. For some communities, barriers to accessing services might understandably include distrust or fear of formal systems, including law enforcement and social services (e.g., fear of criminal justice responses among Latinx immigrants due to the criminalization of immigration into the United States [24, 25]). Communities also might have different conceptualizations of what constitutes IPV, what roles must be assumed by men and women within the family, and when help-seeking is appropriate. In turn, such diversities might affect if, how, and when individuals seek help for IPV [26, 27].

Our results show that preventable IPV-related fatalities is an important contributor to racial and gender-based life expectancy gaps. Regrettably, few programs geared toward preventing IPV before it starts have shown promise in preventing IPV perpetration behaviors [28–30], and evaluation findings from abuser intervention programs for preventing recidivism among those who have perpetrated IPV are mixed and discouraging [31, 32]. Targeting interventions to specific groups of individuals who are more likely to suffer from IPV and die younger may be a helpful strategy. Additionally, the fact that the majority of corollary victims are male should be explored further in future research to determine why corollary victims tend to be male to, in turn, better inform prevention interventions. Future research could also examine sex/race group differences by circumstantial details such as weapons used, circumstances of corollary victim deaths by age, demographic characteristics of suspects, the proportion of PYLL attributed directly to various forms of IPV-related fatality, and economic costs resulting specifically from IPV-related fatalities. In-depth characterization of circumstantial details and perpetrator characteristics could help shed light on both shared and unique needs of various groups at risk of being killed due to IPV. Such work could also lead to more effective, targeted interventions aimed at reducing life-expectancy disparities resulting from IPV.

## Strengths and limitations

A key strength of this study was the NVDRS data, which helped identify IPV-related fatalities across various groups of people. We also used multiple NVDRS variables and narrative review to reliably identify the most IPV-related fatalities possible for data analysis. However, this study includes findings from only 16 states, and some NVDRS cases may have been misclassified as IPV (or not) within the NVDRS. In addition, some IPV-related fatalities, in particular, single suicides (i.e., suicides that were not part of a homicide-suicide) that resulted from IPV, were not included in the current study sample due to limitations in methods for reliably identifying these deaths across 16 states using NVDRS data. Methods to reliably identify single suicides related to IPV are in development and will be used in the future to extend the current study [33, 34]. Considering potential data errors in the NVDRS and narrative review as well as

the exclusion of single suicides, this study underestimates PYLL due to IPV. Additionally, we did not directly control for state-level population and racial and ethnic distributions as a fixed effect in our models; however, we controlled for state-level variation by considering state as a random effects variable. Last, this study could not describe gender non-conforming individuals' experiences with IPV-related fatality due to lack of such information in the data or address relationship configuration (e.g., male on male, female on female violence), nor was the study able to take into account other potentially important individual-level (e.g., socioeconomic status) or community-level (e.g., poverty level) indicators that might help explain the observed disparities, which are important avenues for future research.

We used one well-established method for calculating PYLL which involves using a standard life expectancy for all victims [11–13]. However, disparities in average US life expectancy across sexes and races/ethnicities may lead some researchers to suggest using sex and race/ethnicity-specific life expectancies in PYLL calculations. Existing life expectancy differences, in fact, highlight the inequity in our society based on race, ethnicity, and gender, and using that inequity in PYLL calculation only compounds the inequity by altering the expectation of how long a human should live based on race or gender. If we were to calculate PYLL with sex/race specific average life expectancies, our findings would change—a change that would obscure life expectancy disparities. Accordingly, we used a single, commonly used average life expectancy regardless of sex or race/ethnicity, which is also a well-accepted method [12, 13, 15].

## Conclusions

IPV-related fatalities among intimate partners and corollary victims are costly to society, and the burden of that cost is disproportionately higher among some sex and racial/ethnic groups than others in the 16 US states included in this study. Future research on IPV-related fatalities should consider interrelated effects of sex and race/ethnicity, and seek to explain why particular groups suffer more of the burden of these fatalities. Such research will inform ongoing efforts to prevent IPV and IPV-related fatalities among diverse communities as well as reduce life-expectancy disparities resulting from IPV.

## Acknowledgments

The Centers for Disease Control and Prevention provided the National Violent Death Reporting System data, and individual US states are the data sources. All analyses, findings, interpretations, and conclusions that appear in this article were reached by the authors and do not represent the official position of the Centers for Disease Control and Prevention.

## Author Contributions

**Conceptualization:** Laurie M. Graham, Shabbar I. Ranapurwala, Rebecca J. Macy.

**Data curation:** Laurie M. Graham.

**Formal analysis:** Laurie M. Graham.

**Methodology:** Laurie M. Graham, Shabbar I. Ranapurwala, Catherine Zimmer.

**Project administration:** Laurie M. Graham.

**Supervision:** Shabbar I. Ranapurwala, Catherine Zimmer, Rebecca J. Macy.

**Writing – original draft:** Laurie M. Graham, Shabbar I. Ranapurwala, Sandra L. Martin.

**Writing – review & editing:** Laurie M. Graham, Shabbar I. Ranapurwala, Catherine Zimmer, Rebecca J. Macy, Cynthia F. Rizo, Paul Lanier, Sandra L. Martin.

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
