## [Decision Letter · Decision Letter 0]

9 Oct 2020

PONE-D-20-21187

Disparities in potential years of life lost due to intimate partner violence: Data from 16 states for 2006–2015

PLOS ONE

Dear Dr. Laurie M. Graham,

Thank you for submitting your manuscript to PLOS ONE. After careful consideration, we feel that it has merit but does not fully meet PLOS ONE’s publication criteria as it currently stands. Therefore, we invite you to submit a revised version of the manuscript that addresses the points raised during the review process.

We look forward to receiving your revised manuscript.

Kind regards,

Xiangming Fang

Academic Editor

PLOS ONE

"We would like to thank the Caroline H. and Thomas S. Royster Fellowship awarded by the University of North

Carolina at Chapel Hill Graduate School and Injury and Violence Prevention Fellowship

awarded by the UNC Injury Prevention Research Center which supported, in part, LMG’s time

and effort on this research by supporting her pursuit of a doctoral degree.".

i) We note that you have provided funding information that is not currently declared in your Funding Statement. However, funding information should not appear in the Acknowledgments section or other areas of your manuscript. We will only publish funding information present in the Funding Statement section of the online submission form.

ii) Please remove any funding-related text from the manuscript and let us know how you would like to update your Funding Statement. Currently, your Funding Statement reads as follows:

 "The authors received no specific funding for this work.".

 iii) Please include your amended statements within your cover letter; we will change the online submission form on your behalf.

Reviewers' comments:

Reviewer's Responses to Questions

**Comments to the Author**

1. Is the manuscript technically sound, and do the data support the conclusions?

Reviewer #1: Yes

Reviewer #2: Yes

2. Has the statistical analysis been performed appropriately and rigorously? 

Reviewer #1: Yes

Reviewer #2: Yes

3. Have the authors made all data underlying the findings in their manuscript fully available?

Reviewer #1: Yes

Reviewer #2: Yes

4. Is the manuscript presented in an intelligible fashion and written in standard English?

Reviewer #1: Yes

Reviewer #2: Yes

5. Review Comments to the Author

Reviewer #1: This manuscript, “Disparities in Potential Years of Life Lost Due to Intimate Partner Violence: Data from 16 States for 2006-2015” provides new data on the disproportionate burden of intimate partner violence (IPV)-related fatalities by calculating potential years of life lost (PYLL) by sex and race/ethnicity. The manuscript is well written and contributes to the literature by expressing the burden of IPV-related fatalities beyond incidence rates and by including corollary victims.

However, the manuscript could provide more context for the results of this study by relating it to what is already known about the disproportionate burden of IPV-related deaths. There are some clarifications and considerations that could improve the paper, addressed below.

Title: Appropriate

Abstract: Appropriate

Introduction:

• Provide more statistics for the current incidence/trends of IPV-related deaths by race and sex.

• This paper does a good job explaining or providing footnotes on each category. Considering adding or clarifying if the suspect is included in the count for IPH-suicide.

Methods:

• Were any other covariates considered or limited by data sources?

• Does this analysis account for regional differences?

Results:

• Clarify in Table 2 if “Suspect was law enforcement” means that law enforcement caused the death. It is a little

confusing in that there are potentially two suspects, the one who caused the death and the IPV perpetrator.

• The tables and footnotes are very helpful to see the results laid out.

Discussion:

• If appropriate, perhaps point out any known or informed conclusion that the impact of social distancing could have on

IPV-related fatalities.

• Consider including a comparison example of another disease or issues’ PYLL so the reader can have a general gauge on the total PYLL from IPV.

• I appreciate the notes on disparities of life expectance by race and how it relates to your findings and the limitation in regards to non-binary genders.

• Discuss how generalizable these results are or aren't and how resource allocation and targeted IPV prevention strategies should also explore the paper’s results stratified by region.

• It might be worth bringing up some research on implicit bias toward IPV and likelihood of arrests/receiving support which could prevent fatalities. For instance if the victim is white, the suspect is more likely to be arrested. (McCormack PD, Hirschel D. Race and the Likelihood of Intimate Partner Violence Arrest and Dual Arrest. Race and Justice. September 2018. doi:10.1177/2153368718802352)

Reviewer #2: General Review of the Paper:

The article “Disparities in potential years of life lost due to intimate partner violence: data from 16 states for 2006-2015” makes a significant contribution to the field of public health by exploring the societal costs of intimate partner violence (IPV). The literature review was well-written and sufficiently identified the gaps in existing IPV research. The focus on disparities as it related to both the number of IPV-related deaths and in the potential years of life lost due to IPV-related deaths is an innovative focus and significant contribution to the literature. The analyses also included analysis of perpetrator-victim relationships, which is important and informative for prevention programming. Limitations of the research included the use of a general life expectancy in order to address disparities, despite knowledge of disparities in life expectancy amongst racial groups; however, the authors offered a well-justified explanation for this decision. Additional feedback for the authors according to manuscript section is provided below.

TITLE AND ABSTRACT

• The title is appropriate.

• The abstract was comprehensive. One additional consideration might be the addition of the analyses documenting the victim -suspect relationship by fatality types.

INTRODUCTION

• This section summarizes the current literature and identifies the gaps in current research. It is well written and does a nice job justifying the research questions.

• Line 74-75: Are there any targeted prevention efforts that have been successful, or would these need to be developed and tested based on study results. Add a bit more clarification to this statement.

METHODS

• This section was well-written. There is one point that deserves more clarification (lines 121-125). Was a pre-existing NVDRS variable used or was a new variable created through narrative review (or a mixture of both).

RESULTS

• Clear and informative

DISCUSSION

• Note as a limitation that the differences in state density and racial and ethnic group composition among the 16 NVDRS states were not accounted for in analyses.

• Researchers should consider briefly mentioning other individual (SES, Education) and community level (crime, poverty level)variables that impact disparities that could be explored in future research.

• In reference to corollary victims being male more often than female, researchers might consider adding a statement about how young boys are subjected to physical child abuse more than girls.

o A future study direction about exploring the ages of corollary victims could also be added.

TABLES/FIGURES

• The tables were generally informative and comprehensive.

• For Table 1, the inclusion of full-dataset number and percentages seemed unnecessary since the analysis was based on the reduced dataset.

6. PLOS authors have the option to publish the peer review history of their article (what does this mean?). If published, this will include your full peer review and any attached files.

Reviewer #1: **Yes: **Jyll Walsh

Reviewer #2: No

---

## [Author Response · Author response to Decision Letter 0]

23 Nov 2020

The authors of this manuscript acknowledge and thank the reviewers for the time they spent reviewing this manuscript and offering thoughtful comments. We appreciated the positive comments made about the manuscript, and the critical comments helped us see ways in which we could strengthen our manuscript and research. We reviewed the recommendations and revised the manuscript in response the specific feedback provided by reviewers (see edits made in the manuscript in track changes). Additionally, we provided our detailed responses to reviewer comments in the table included as the "Response to Reviewers" document.

---

## [Decision Letter · Decision Letter 1]

20 Jan 2021

Disparities in potential years of life lost due to intimate partner violence: Data from 16 states for 2006–2015

PONE-D-20-21187R1

Dear Dr. Graham,

We’re pleased to inform you that your manuscript has been judged scientifically suitable for publication and will be formally accepted for publication once it meets all outstanding technical requirements.

Kind regards,

Xiangming Fang

Academic Editor

PLOS ONE

Reviewers' comments:

Reviewer's Responses to Questions

**Comments to the Author**

1. If the authors have adequately addressed your comments raised in a previous round of review and you feel that this manuscript is now acceptable for publication, you may indicate that here to bypass the “Comments to the Author” section, enter your conflict of interest statement in the “Confidential to Editor” section, and submit your "Accept" recommendation.

Reviewer #1: All comments have been addressed

Reviewer #2: All comments have been addressed

2. Is the manuscript technically sound, and do the data support the conclusions?

Reviewer #1: Yes

Reviewer #2: Yes

3. Has the statistical analysis been performed appropriately and rigorously? 

Reviewer #1: Yes

Reviewer #2: Yes

4. Have the authors made all data underlying the findings in their manuscript fully available?

Reviewer #1: Yes

Reviewer #2: Yes

5. Is the manuscript presented in an intelligible fashion and written in standard English?

Reviewer #1: Yes

Reviewer #2: Yes

6. Review Comments to the Author

Reviewer #1: The authors appropriately addressed all the comments I had submitted within the article or in the review commentary. I have no further edits for this article.

Reviewer #2: (No Response)

7. PLOS authors have the option to publish the peer review history of their article (what does this mean?). If published, this will include your full peer review and any attached files.

Reviewer #1: No

Reviewer #2: No

---

## [Editor Report · Acceptance letter]

27 Jan 2021

PONE-D-20-21187R1 

Disparities in potential years of life lost due to intimate partner violence: Data from 16 states for 2006–2015 

Dear Dr. Graham:

I'm pleased to inform you that your manuscript has been deemed suitable for publication in PLOS ONE. Congratulations! Your manuscript is now with our production department. 

Kind regards, 

on behalf of

Dr. Xiangming Fang 

Academic Editor

PLOS ONE